# Inflammation and hemostasis in loiasis before and after ivermectin treatment: A biological pilot cross-sectional study

Tristan M. Lepage[1,2], Narcisse Nzune-Toche[3], Lucie A. Nkwengoua[4], Hugues C. Nana-Djeunga[3], Sebastien D. S. Pion[2], Joseph Kamgno[3,4], Charlotte Boullé[1,2], Jérémy T. Campillo[2], Michel Boussinesq[2], Claude T. Tayou[4,5], Cédric B. Chesnais[2]*

1 Department of Infectious and Tropical Diseases, Montpellier University Teaching Hospital, Montpellier, France, 2 TransVIHMI, Montpellier University, INSERM Unité 1175, Institut de Recherche pour le Développement (IRD), Montpellier, France, 3 Higher Institute of Scientific and Medical Research (ISM), Yaoundé, Cameroon, 4 Faculty of Medicine and Biomedical Sciences, University of Yaoundé I, Yaoundé, Cameroon, 5 Hematology laboratory, Yaoundé University Teaching Hospital, Yaoundé, Cameroon

* cedric.chesnais@ird.fr

**Data Availability Statement:** Data are available from the DataSuds repository (IRD, France) at https://doi.org/10.23708/7DIZI5 for researchers

## Abstract

We assessed the impact of loiasis and its treatment with ivermectin on hemostasis and inflammation in 38 adults in Cameroon. Participants were divided into four balanced groups based on their *Loa loa* microfilarial densities. At baseline, eosinophils and platelets increased with microfilarial densities (p<0.001 and p = 0.044, respectively). At day 4 following ivermectin administration, median eosinophils rose from 171/μL to 1,095/μL (p<0.001) and median D-dimers increased from 582 ng/mL to 745 ng/mL (p = 0.024) with a median relative variation of +34.7%. C-reactive protein, fibrinogen, and alpha-1-globulin also increased significantly after treatment. Ivermectin treatment appeared to induce inflammation, coagulation activation and fibrinolysis.

## Author summary

Loiasis, a parasitic disease endemic to Central Africa, is caused by *Loa loa*, a worm that releases embryos called microfilariae into the bloodstream. Although historically considered a benign disease, loiasis can significantly shorten life expectancy in individuals with high microfilarial densities, and may lead to severe complications following treatment with ivermectin, a widely used antiparasitic drug. Several studies suggest that a potential dysregulation of inflammation and blood clotting may contribute to these complications.

In this study, we analyzed blood samples from individuals with varying levels of loiasis infection. We found that those with higher microfilarial densities had elevated eosinophil and platelet counts, indicating immune activation. After ivermectin treatment, we observed significant changes in several biological markers, including eosinophil counts, C-reactive protein, and D-dimers. These changes suggest an increase in inflammation and activation of blood clotting mechanisms following treatment.

**Funding:** This work was supported by the European Research Council (ERC, https://erc.europa.eu) under the European Union's Horizon 2020 research and innovation program [grant agreement No 949963]. C.B.C. is the carrier of this grant. The funders had no role in the design and conduct of the study; collection, management, analysis, and interpretation of the data; preparation, review, or approval of the manuscript; and decision to submit the manuscript for publication.

**Competing interests:** The authors have declared that no competing interests exist.

This study provides new insights into the biological effects of loiasis and its treatment, and our findings may help explain the severe complications observed in some individuals treated with ivermectin.

## Background

Loiasis, caused by the filarial nematode *Loa loa* and transmitted from human to human by tabanids, is exclusively endemic in central Africa. In infected individuals, adult female worms release embryos called microfilariae (mfs) which circulate in the bloodstream according to a diurnal cycle, with densities occasionally surpassing 100,000 mfs per mL of blood (mf/mL).

Although loiasis is historically considered as a benign disease, recent findings suggest that microfilaremic individuals have a significantly shortened life expectancy [1]. In addition, studies in the Republic of Congo showed that high *L. loa* microfilarial density (MFD) was significantly associated with proteinuria [2], cognitive impairment [3], and increased vascular stiffness [4].

Eosinophilia exceeding 5,000 /μL is common among subjects with loiasis. Eosinophils, known to express tissue factor, may trigger blood coagulation [5], leading to thrombosis. Reports indicate that arterial and venous thrombotic events can occur in patients with parasite-related hypereosinophilia [6], with documented instances of thrombosis [7] and micro-emboli formation in capillary vessels [8], occurring both spontaneously and as part of post-ivermectin (IVM) *Loa*-related serious adverse events (SAEs). Indeed, IVM mass drug administration in onchocerciasis endemic regions may cause SAEs such as life-threatening encephalopathies in patients with high *L. loa* MFD above 30,000 mf/mL, due to its potent microfilaricidal effect [9]. Evidence from primate models suggests an association between post-IVM encephalopathy and hemostatic dysregulation, characterized by degenerating mfs in small vessels associated with fibrin deposition, endothelial damages, and hemorrhages [10]. Similarly, humans experiencing post-ivermectin encephalopathy often exhibit conjunctival and retinal hemorrhages [11].

A potential dysregulation of hemostasis may contribute to the excess morbidity and mortality associated with loiasis and its treatment with IVM. However, data on this phenomenon remains limited and unexplored in humans. The objective of the present study was to assess for the first time the impact of both *L. loa* MFD and IVM on human hemostasis and inflammation blood parameters.

## Methods

### Ethics statement

The study adhered to ethical guidelines outlined in the Declaration of Helsinki, version 13, and was approved by the Ethics Committee for Human Research of the Catholic University of Central Africa (N° 2022/022503/CEIRSH/ESS/MH) and by Yaoundé University Teaching Hospital (N° 4181/AR/CHUY/DG/DGA/CAPRC). Only individuals who provided informed consent were enrolled and assigned an individual code for data recording. Blood parameters results were shared with all participants.

### Study design and participants

The study was conducted from July to December 2022 in two Health Districts of the Center Region of Cameroon (Mbalmayo and Awaé) where loiasis is highly endemic. Participants

consisted of consenting adults aged 25 to 70 who had been screened for *L. loa* mfs. Exclusion criteria included pregnancy, concurrent pro-inflammatory conditions (such as infection, cancer or hematological disorder), current use of anticoagulant or anti-inflammatory medications, or known allergy to IVM. Eligible participants were stratified into 4 groups based on their *L. loa* MFD: 0 mf/mL; 20–4,999 mf/mL; 5,000–19,999 mf/mL; and 20,000–40,000 mf/mL.

Venous blood samples were collected from all groups at baseline (D0) to measure hematological, hemostatic, and biochemical parameters. Subsequently, individuals in the 20–4,999 mf/mL and 5,000–19,999 mf/mL groups received a single oral dose of IVM (150μg/kg), and the same blood parameters were reassessed four days after (D4) treatment.

## Sample size

No data exists on the relationships between loiasis and hemostasis. In a study evaluating D-dimer levels in dogs infected with another filaria (*Dirofilaria immitis*) [12], the proportion of elevated D-dimers (>0.2 μg/ml) was 48% (22/46) in microfilaremic dogs, compared to 0% in amicrofilaremic dogs. Based on this finding, the inclusion of 16 *L. loa* microfilaremic subjects and 16 amicrofilaremic subjects would provide an 80% power with an alpha risk of 5% to detect such a difference in loiasis. However, it is difficult to extrapolate these assumptions to loiasis, which is a different disease with a different host. Therefore, we planned to include 10 *L. loa* amicrofilaremic subjects, and 30 microfilaremic subjects in 3 groups of ascending MFD.

## Laboratory analyses

Blood was collected by fingerprick between 10:00 am and 3:00 pm in non-heparinized capillaries and spread on microscope slides to prepare two 50 μL thick blood smears which were dried, dehemoglobinized, and stained with Giemsa within 24 hours. *L. loa* mfs present on slides were counted using a microscope at 100-fold magnification at the Higher Institute for Scientific and Medical Research in Yaoundé, and averages of both slides were multiplied by 20 to express MFD per milliliter of blood.

Venous blood samples were collected from each participant using EDTA, sodium citrate, and dry blood tubes. Complete blood counts (flow cytometry), erythrocyte sedimentation rate (ESR) after one-hour (Westergren method), C-reactive protein (CRP) levels assessed by nephelometry, prothrombin time (PT), activated thromboplastin time (aPTT) and fibrinogen levels (chronometry) were measured at the hematology laboratory of the Yaoundé University Teaching Hospital. Wright-stained blood smears were prepared, and the percentage of eosinophils was recorded after examining 200 leukocytes. Eosinophil counts were calculated by multiplying the percentage of eosinophils by the white blood cells count.

Serum protein electrophoresis, ferritin (chemiluminescence) and D-dimers (ELISA) were assessed at the Centre Pasteur in Yaoundé. Finally, prothrombin fragment F1+2 (ELISA) and plasminogen activity (amidolytic method followed by nephelometry) were evaluated in a subset of participants (10 with no microfilariae and 10 with 20–40,000 mf/mL) at the Cerba Laboratory in France.

## Statistical analysis

Hematological, hemostatic, and biochemical blood parameters were compared among the four MFD groups at baseline using the Kruskal-Wallis test. When the test yielded a p-value < 0.1, a Cuzick's test for trend was performed. Since D-dimers levels can be influenced by age and sex, the relationship between D-dimer and MFD was also assessed using a linear regression, with adjustments on age and sex. Prothrombin fragment F1+2 and plasminogen, assessed in a subset of 20 participants, were compared between amicrofilaremic and

microfilaremic subjects using the Wilcoxon rank-sum test. Finally, within the groups of individuals treated with IVM (20–4,999 mf/mL and 5,000–19,999 mf/mL), changes in hematological, hemostatic, and biochemical blood parameters were compared between baseline and D4 post-IVM using a Wilcoxon signed-rank test for paired observations.

## Results

Twenty males and 18 females aged between 28 and 67 years (median age: 43 years old) participated in the study.

Table 1 presents the blood counts and the inflammation and hemostasis parameters according to the participants' *L. loa* MFD categories. There was a trend towards increased leucocyte, granulocyte and eosinophil counts with increasing MFD (p = 0.011, p = 0.013 and p<0.001, respectively, Cuzick's tests for trend). Median eosinophil counts were significantly higher in patients with a MFD exceeding 20,000 mf/mL compared to those with no mf (1,016/µL vs 111/µL, p<0.001). Similarly, median platelet counts were also significantly higher in the same group (254,500/µL vs 180,000/µL, p = 0.044). Inflammation and hemostasis parameters did not significantly differ between groups. Linear regression analysis of D-dimers by MFD, adjusted on age and sex, did not show any significant associations. Median ESR were consistently increased across all groups (normal values: < 20 mm/h). Gamma-globulins were also elevated in all groups, with medians from 19 to 23 g/L (normal values: < 11 g/L). Clonal gamma-globulin expression was suspected in the serum protein electrophoresis in 3 patients, each in separate groups.

Eighteen individuals with an MFD between 20 and 19,999 mf/mL received a single dose of IVM and were re-examined at D4 (Table 2). Post-treatment, the median MFD significantly decreased from 3,605 mf/mL to 1,810 mf/mL. Median eosinophil counts rose from 171/µL to 1,095/µL (p<0.001). Various inflammation markers changed significantly: CRP, ESR, fibrinogen and alpha-1-globulin levels increased (p = 0.038, p = 0.010, p = 0.002 and p = 0.044, respectively), while albumin level decreased p = 0.044). Among hemostatic parameters, median D-dimers significantly increased, from 582 ng/mL to 745 ng/mL after IVM (p = 0.024) with a median relative variation of +34.7% (interquartile range: 1.44–72.5%). Prothrombin fragment 1+2 did not significantly vary (252 versus 469 pmol/L, p = 0.313). No new clonal gamma-globulin expression on protein electrophoresis was suspected, while previously identified clonal expression in treated patients did not change after IVM.

## Discussion

In this pilot biological study, we describe for the first time hemostasis and inflammation parameters associated with loiasis. Our results show a remarkable trend towards increased leukocytes and granulocytes, mainly driven by elevated eosinophils, with increasing L. loa MFD. In a 2022 biological study in Gabon [13], eosinophil counts were also positively correlated with *L. loa* MFD, with an estimated increase in eosinophil counts every 10-fold increase in parasitemia (p-adj. = 0.012, ß-estimate: 0.17[0.04–0.31]). Chronic eosinophilia, a characteristic feature of loiasis, is known for its ability to trigger severe health conditions, such as endomyocardial fibrosis, tropical pulmonary eosinophilia, and thromboembolic events, consequently increasing the risk of mortality associated with loiasis [1].

The notable increase in platelets with increasing MFDs suggests their probable role in filarial infections. Prior research has established that platelets mediate the immunological killing of *L. loa* mfs [14,15], while data in lymphatic filariasis also demonstrated mfs' ability to modulate platelet functions [16].

**Table 1. Blood count, inflammation, and hemostasis parameters according to *L. loa* microfilaremia.**

| Characteristic | 0<br>mf/mL, N = 10 | 20–4,999<br>mf/mL, N = 11 | 5,000–19,999<br>mf/mL, N = 7 | 20,000–40,000<br>mf/mL, N = 10 | p-value |
|---|---|---|---|---|---|
| Sex[1] | | | | | |
| Male | 5 (50%) | 6 (55%) | 3 (43%) | 4 (40%) | |
| Female | 5 (50%) | 5 (45%) | 4 (57%) | 6 (60%) | |
| Age (years)[2] | 40 (38, 52) | 44 (41, 60) | 38 (38, 41) | 54 (43, 58) | |
| *Blood count* | | | | | |
| Hemoglobin (g/dl)[2] | 12.60 (12.13, 13.00) | 12.90 (11.25, 13.35) | 13.10 (11.45, 14.85) | 11.70 (11.10, 12.58) | 0.681[†] |
| Leucocytes (/µl)[2] | 5,535 (4,770, 6,023) | 6,440 (5,315, 8,460) | 7,160 (6,400, 7,405) | 8,825 (6,410, 9,675) | 0.073[†] |
| Granulocytes (/µl)[2] | 2,300 (2,168, 3,793) | 3,320 (3,055, 5,405) | 3,670 (3,560, 4,215) | 5,680 (2,748, 6,635) | 0.078[†] |
| Eosinophils (/µl)[2] | 111 (96, 121) | 129 (107, 199) | 330 (198, 454) | 1,016 (273, 1,806) | **<0.001**[†] |
| Eosinophils (%)[2] | 2.0 (2.0, 2.0) | 2.0 (2.0, 2.0) | 4.0 (2.5, 6.5) | 10.5 (4.5, 18.0) | **<0.001**[†] |
| Lymphocytes (/µl)[2] | 2,185 (1,795, 2,530) | 2,540 (1,605, 2,965) | 2,320 (2,205, 2,460) | 2,615 (2,325, 2,980) | 0.451[†] |
| Platelets (/µl)[2] | 180,000 (163,250, 317,750) | 180,000 (116,500, 206,500) | 241,000 (217,000, 255,500) | 254,500 (198,500, 277,250) | **0.044**[†] |
| *Inflammation* | | | | | |
| CRP (mg/l)[2] | 4 (3, 8) | 3 (3, 6) | 5 (4, 9) | 4 (3, 7) | 0.671[†] |
| ESR (mm/h)[2] | 53 (38, 85) | 53 (33, 82) | 58 (47, 98) | 70 (39, 91) | 0.791[†] |
| Fibrinogen (g/l)[2] | 2.70 (2.32, 2.91) | 2.50 (2.30, 3.22) | 2.54 (2.45, 3.54) | 2.52 (2.35, 3.12) | 0.710[†] |
| Ferritin (µg/l)[2] | 131 (59, 269) | 213 (157, 284) | 269 (113, 350) | 250 (127, 357) | 0.569[†] |
| Albumin (g/l)[2] | 47 (40, 54) | 40 (38, 42) | 45 (38, 52) | 51 (43, 53) | 0.258[†] |
| Alpha-1-Globulin (g/l)[2] | 1.40 (1.10, 3.25) | 1.80 (1.70, 1.88) | 1.30 (1.15, 3.05) | 1.20 (1.00, 2.88) | 0.791[†] |
| Alpha-2-Globulin (g/l)[2] | 6.25 (4.45, 6.60) | 5.85 (4.48, 6.78) | 5.40 (4.70, 5.70) | 5.05 (4.63, 6.28) | 0.763[†] |
| Beta-Globulin (g/l)[2] | 6.60 (5.20, 8.10) | 6.75 (6.35, 7.30) | 6.50 (6.05, 8.25) | 5.85 (5.25, 7.60) | 0.851[†] |
| Gamma-Globulin (g/l)[2] | 19 (16, 28) | 23 (19, 28) | 21 (17, 25) | 19 (17, 26) | 0.664[†] |
| *Hemostasis* | | | | | |
| PT (%)[2] | 86 (79, 95) | 98 (81, 100) | 85 (82, 88) | 87 (77, 91) | 0.502[†] |
| aPTT ratio[2] | 0.95 (0.90, 0.96) | 0.92 (0.87, 1.05) | 0.98 (0.96, 1.05) | 0.93 (0.85, 1.09) | 0.420[†] |
| D-Dimers (ng/ml)[2] | 561 (395, 1,005) | 587 (307, 858) | 577 (382, 624) | 634 (460, 1,223) | 0.765[†] |
| F1+2[*] (pmol/l)[2] | 342 (252, 390) | 262 (252, 302) | | | 0.438[‡] |
| Plasminogen (%)[2] | 100 (89, 118) | 100 (96, 110) | | | 0.970[‡] |

[1] n (%)

[2] Median (IQR)

[*] Prothrombin fragment 1+2

[†] Kruskal-Wallis rank sum test

[‡] Wilcoxon rank sum test

Reference ranges: Hemoglobin 12,0–17,4 g/dL; Leucocytes 5,000–10,000/µL; Granulocytes 2,500–7,500/µL; Eosinophils < 500/µL; Lymphocytes 1,300–4,000/µL; Platelets 150,000–400,000/µL; CRP < 5 mg/L; ESR < 20 mm/h; Fibrinogen 2.0–4.0 g/L; Ferritin 18–370 ng/mL; Albumin 43–51 g/L; Alpha-1-Globulin 1.0–2.0 g/L; Alpha-2-Globulin 5.0–8.0 g/L; Beta-Globulin 6.0–9.0 g/L; Gamma-Globulin 6.0–11.0 g/L; PT > 70%; aPTT ratio 0.8–1.2; D-Dimers < 500 ng/mL; F1+2 69–229 pmol/L; Plasminogen 75–140%.

Furthermore, our study highlighted a polyclonal elevation of gamma-globulins across all MFD groups, suggesting a systemic immune response. This phenomenon is consistent with a previous study in Ghana, were 75% of participants had gamma-globulins > 16 g/L [17]. This hypergammaglobulinemia might be explained by the influence of genetic as well as environmental factors. ESR were increased in all groups, possibly driven by hypergammaglobulinemia, which is known to accelerate erythrocyte sedimentation.

Interestingly, while inflammation and hemostasis parameters did not significantly vary with *L. loa* MFD at baseline, treatment with IVM resulted in significant alterations in several

**Table 2. Blood count, inflammation, and hemostasis parameters at baseline and 4 days after IVM treatment in patients with *L. loa* microfilaremia.**

| Characteristic | Baseline, N = 18 | D4 after IVM, N = 18 | p-value[2] |
|---|---|---|---|
| MFD* (mf/mL) [1] | 3,605 (1,645, 10,020) | 1,810 (545, 2,933) | |
| *Blood count* | | | |
| Hemoglobin (g/dl) [1] | 12.90 (11.08, 14.00) | 11.80 (9.65, 13.60) | 0.058 |
| Leucocytes (/μl) [1] | 6,600 (5,853, 7,460) | 7,070 (5,975, 9,598) | 0.119 |
| Granulocytes (/μl) [1] | 3,635 (3,138, 4,758) | 4,010 (2,165, 6,253) | 0.417 |
| Eosinophils (/μl) [1] | 171 (121, 239) | 1,095 (281, 2,179) | <**0.001** |
| Eosinophils (%)[1] | 2 (2, 3) | 20 (5, 31) | <**0.001** |
| Lymphocytes (/μl) [1] | 2,460 (2,103, 2,648) | 2,385 (1,723, 3,218) | 0.663 |
| Platelets (/μl) [1] | 213,000 (170,250, 240,750) | 187,000 (156,750, 217,750) | 0.170 |
| *Inflammation* | | | |
| CRP (mg/l) [1] | 5 (3, 6) | 10 (3, 15) | **0.038** |
| ESR (mm/h) [1] | 56 (40, 92) | 81 (60, 97) | **0.010** |
| Fibrinogen (g/l) [1] | 2.52 (2.35, 3.58) | 3.14 (2.63, 3.76) | **0.002** |
| Ferritin (μg/l) [1] | 234 (138, 337) | 224 (120, 366) | 0.089 |
| Albumin (g/l) [1] | 41 (38, 47) | 39 (36, 44) | **0.044** |
| Alpha-1-Globulin (g/l) [1] | 1.80 (1.30, 2.80) | 1.95 (1.73, 3.48) | **0.044** |
| Alpha-2-Globulin (g/l) [1] | 5.40 (4.60, 6.50) | 5.80 (5.08, 6.60) | 0.255 |
| Beta-Globulin (g/l) [1] | 6.60 (6.20, 7.90) | 6.70 (6.23, 8.25) | 0.842 |
| Gamma-Globulin (g/l) [1] | 21 (19, 26) | 22 (19, 26) | 0.831 |
| *Hemostasis* | | | |
| PT (%)[1] | 88 (81, 100) | 96 (86, 100) | 0.220 |
| aPTT ratio[1] | 0.97 (0.92, 1.05) | 0.99 (0.90, 1.08) | 0.257 |
| D-Dimers (ng/ml)[1] | 582 (352, 820) | 745 (512, 1,061) | **0.024** |
| F1+2† (pmol/l) [1] | 252 (242, 290) (N = 6) | 469 (334, 529) (N = 7) | 0.313 |
| Plasminogen (%)[1] | 100 (96, 107) (N = 7) | 90 (85, 102) (N = 6) | 0.438 |

[1] Median (IQR)

[2] Wilcoxon signed-rank test

* Microfilarial density

†Prothrombin fragment 1+2

Reference ranges: Hemoglobin 12,0–17,4 g/dL; Leucocytes 5,000–10,000/μL; Granulocytes 2,500–7,500/μL; Eosinophils < 500/μL; Lymphocytes 1,300–4,000/μL; Platelets 150,000–400,000/μL; CRP < 5 mg/L; ESR < 20 mm/h; Fibrinogen 2.0–4.0 g/L; Ferritin 18–370 ng/mL; Albumin 43–51 g/L; Alpha-1-Globulin 1.0–2.0 g/L; Alpha-2-Globulin 5.0–8.0 g/L; Beta-Globulin 6.0–9.0 g/L; Gamma-Globulin 6.0–11.0 g/L; PT > 70%; aPTT ratio 0.8–1.2; D-Dimers < 500 ng/mL; F1+2 69–229 pmol/L; Plasminogen 75–140%.

inflammatory markers (such as CRP, ESR, fibrinogen, albumin and alpha-1-globulins). While post-IVM CRP increase has been reported previously [18], we document for the first time significant changes in a range of inflammatory parameters. We observed a marked increase in eosinophils, consistent with previous reports of interleukin-5 (IL-5)-driven eosinophilia following microfilarial killing by IVM [19].

The increase in D-dimers observed on D4 post-IVM suggests both enhanced fibrinolysis and coagulation activation which happen simultaneously. This could be attributed to the close relationship between hemostasis and inflammation, where inflammation induces fibrin formation in the extravascular space, facilitating immune cells functions [20]. Consequently, secondary fibrinolysis generates D-dimers, which correlate with inflammation severity. However, this correlation is modest, as previous studies found a Spearman's correlation coefficient of 0.18 between CRP and D-dimers, indicating a limited response of D-dimers to CRP elevation [21].

Therefore, in addition to inflammation, the potential involvement of thrombotic processes in the increase of D-dimers after IVM cannot be ruled out, potentially both driven by intravascular dying mfs and inflammation-induced hypercoagulable condition.

In addition to previous reports of post-IVM retinal emboli [11], this hypothesis is supported by autopsies of post-IVM encephalopathies in baboon models, which show abnormalities ranging from intravascular degenerating mfs and vessel wall infiltration by immune cells such as eosinophils, to fibrin deposition and endothelial disruption with extravascular hemosiderin deposits [10,22]. These findings provide a plausible pathological basis for the observed post-IVM treatment biological changes, potentially contributing to the pathogenesis of post-IVM SAEs.

Despite the valuable insights gained, our study has limitations. The small sample size and absence of preexisting data require cautious interpretation of our findings, highlighting the need for larger dedicated studies to validate our observations and understand their clinical implications. However, our results, particularly in baseline comparisons, may guide further research with adequate sample sizes, especially to explore the impact of loiasis on the hemostatic system. Additionally, potential confounding by other health conditions among participants, particularly in a low-resource healthcare setting, warrants consideration. Despite this, hematological changes showed a significant increase trend with *L. loa* MFD, aligning with previous findings [13]. However, as only *L. loa* MFD was assessed, potential confounding by other parasitic infections sensitive to IVM cannot be ruled out. Finally, our reliance on a single post-IVM evaluation timepoint may have limited our ability to detect biological abnormalities occurring outside this timeframe.

In conclusion, this pilot study identified several biological changes potentially associated with loiasis, occurring both spontaneously and after treatment with IVM. Further dedicated studies are needed to confirm these results, which might participate in loiasis excess mortality as well as post-IVM SAEs pathogenesis.

## Acknowledgments

We thank all the participants who agreed to take part in this study as well as the village chiefs and health workers involved.

## Author Contributions

**Conceptualization:** Charlotte Boullé, Michel Boussinesq, Claude T. Tayou, Cédric B. Chesnais.

**Data curation:** Tristan M. Lepage.

**Formal analysis:** Tristan M. Lepage, Sebastien D. S. Pion.

**Funding acquisition:** Cédric B. Chesnais.

**Investigation:** Tristan M. Lepage, Narcisse Nzune-Toche, Lucie A. Nkwengoua.

**Methodology:** Tristan M. Lepage, Charlotte Boullé, Michel Boussinesq, Claude T. Tayou, Cédric B. Chesnais.

**Project administration:** Tristan M. Lepage, Narcisse Nzune-Toche.

**Resources:** Hugues C. Nana-Djeunga, Joseph Kamgno, Claude T. Tayou.

**Software:** Tristan M. Lepage, Sebastien D. S. Pion.

**Supervision:** Tristan M. Lepage, Hugues C. Nana-Djeunga, Joseph Kamgno, Claude T. Tayou, Cédric B. Chesnais.

**Validation:** Michel Boussinesq, Claude T. Tayou, Cédric B. Chesnais.

**Writing – original draft:** Tristan M. Lepage.

**Writing – review & editing:** Tristan M. Lepage, Narcisse Nzune-Toche, Sebastien D. S. Pion, Joseph Kamgno, Charlotte Boullé, Jérémy T. Campillo, Michel Boussinesq, Claude T. Tayou, Cédric B. Chesnais.

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
