## [Decision Letter · Decision Letter 0]

4 Oct 2024

Dear Dr. Lepage,

Thank you very much for submitting your manuscript "Inflammation and fibrinolysis in loiasis before and after ivermectin treatment: a biological pilot cross-sectional study" for consideration at PLOS Neglected Tropical Diseases. As with all papers reviewed by the journal, your manuscript was reviewed by members of the editorial board and by several independent reviewers. In light of the reviews (below this email), we would like to invite the resubmission of a significantly-revised version that takes into account the reviewers' comments. 

We cannot make any decision about publication until we have seen the revised manuscript and your response to the reviewers' comments. Your revised manuscript is also likely to be sent to reviewers for further evaluation.

Sincerely,

Feng Xue, Ph.D.

Guest Editor

Jong-Yil Chai

Section Editor

Reviewer's Responses to Questions

**Key Review Criteria Required for Acceptance?**

**Methods**

-Are the objectives of the study clearly articulated with a clear testable hypothesis stated?

-Is the study design appropriate to address the stated objectives?

-Is the population clearly described and appropriate for the hypothesis being tested?

-Is the sample size sufficient to ensure adequate power to address the hypothesis being tested?

-Were correct statistical analysis used to support conclusions?

-Are there concerns about ethical or regulatory requirements being met?

Reviewer #1: The article does not clearly state why ivermectin therapy was chosen. The introduction should clearly state the specific advantages of this therapy over others.

Authors should indicate whether each type of loiasis treatment is associated with hemostatic disorders, or whether this is specific for the ivermectin therapy only.

It is a bit confusing, why patients with the highest MFD count (20,000-40,000) did not receive the treatment.

Reviewer #2: (No Response)

Reviewer #3: In concerns a pilot study with appropriate design.

**Results**

-Does the analysis presented match the analysis plan?

-Are the results clearly and completely presented?

-Are the figures (Tables, Images) of sufficient quality for clarity?

Reviewer #1: Elevation of d-dimers, observed in this work, is an indirect sign of the hemostasis dysregulation. In order to evaluate the patients’ hemostasis state, global hemostasis test, sensitive to both hypo- and hyper-coagulation, like thrombin generation test, or thrombodynamics, should be assessed.

Reviewer #2: (No Response)

Reviewer #3: -The authors describe an elevation of neutrophils in the results section, however only granulocytes are depicted in table 1. Can the authors add the neutrophil counts here? 

-In the first paragraph of the results section, the authors describe that ‘there was a trend towards increased leucocytes, neutrophils and eosinophils’. However, for eosinophils this is not only a trend, but a highly significant elevation. The text should be adjusted to make this more clear. 

-Platelets are significantly lower in 20-4,999 mf/mL group. Can the authors discuss on these findings?

**Conclusions**

-Are the conclusions supported by the data presented?

-Are the limitations of analysis clearly described?

-Do the authors discuss how these data can be helpful to advance our understanding of the topic under study?

-Is public health relevance addressed?

Reviewer #1: (No Response)

Reviewer #2: (No Response)

Reviewer #3: The conclusion (inflammation activation and pronounced fibrinolysis, indicative of coagulation activation) is mainly based on elevation of neutrophils and eosinophils, as well as D-dimer levels. More specific markers need to be included to be able to conclude this.

**Editorial and Data Presentation Modifications?**

Reviewer #1: (No Response)

Reviewer #2: (No Response)

Reviewer #3: Modifications are mentioned in the other sections.

**Summary and General Comments**

Reviewer #1: (No Response)

Reviewer #2: Lepage et al. report an observational study in patients with loasis, in whom markers of inflammation and coagulation were measured. They observed that eosinophils increased with MFD. In a subgroup that received ivermectin treatment, an increase of eosinophils was also associated with an increase in inflammatory parameters and D-dimer. This is an interesting study, whose results improves the understanding of loasis.

Major comments:

1. I find it difficult to understand, what is the authors’ main interpretation of d-dimer changes. The abstract suggests that it is increased fibrinolysis. “Ivermectin… appeared to induce… pronounced fibrinolysis” (line 29,30). The paragraph about d-dimer changes in the discussion (line 176 ff) also starts with an interpretation of increased d-dimer as an indicator of increased fibrinolysis, then suggests that it might be indicative of increased thrombus formation / hypercoagulability. This is confusing, as a “pronounced fibrinolysis” would imply a bleeding phenotype. D-dimer is a compound biomarker of increased coagulation activation AND fibrinolysis activation which happen simultaneously. It is not a fibrinolysis biomarker, and – in the absence of other findings – is not indicative of thrombotic processes.

2. Laboratory analyses should be described in more detail. Information regarding reagents and analyzers should be provided. The method of fibrinogen measurement should be provided (Clauss method?).

3. Statistical analysis. Were the data tested for normality? Data are presented as mean and SD, but non-parametric tests were used to compare groups. Generally, normally distributed data should be presented as mean and SD, otherwise it is usually presented as median and IQR. Normally distributed data should be compared using parametric tests.

4. Presentation of results. Reference ranges for parameters should be provided in the tables. Age and sex of the study participants in each of the four cohorts should be provided in Table1. In Table 2, MFD at baseline and at D4 after treatment should be provided. The terms neutrophils or granulocytes should be used consistently. Some figures, e.g. box plots, or scatter plots for possible correlations should be considered. 

Minor comments:

5. Abstract. Use of the term “correlation” wrongly suggests Spearman or Pearson analysis, when the observation in fact is, that granulocytes increase with MFD.

6. The introduction should include background information regarding ivermectin.

7. Why did the group with the highest MFD not received ivermectin treatment?

8. Line 144, it should not be stated that F1+2 increased, as the p-value was quite high, and the number of patients (n=7) was small.

Reviewer #3: Tristan Lepage and colleagues have investigated the impact of loiasis and its treatment with ivermectin on hemostasis and inflammation. It concerns a pilot study with a limited number of participants, and differences were observed in some general inflammation and hemostasis parameters. For me, the analyzed parameters are too general and conclusions are drawn by elevation of some of these parameters. 

-Most importantly, coagulation activation is now suggested based on elevated D-dimer levels. Better markers, for examples TAT-complexes, should be provided to be able to demonstrate coagulation activation. 

-Eligible participants for this study, screened for the presence of L. loa mfs, were stratified into 4 groups based on their MFD. However, how is it then possible to have a group with 0 mf/mL? 

-Measuring von Willebrand factor antigen levels would be of added value for this study to demonstrate a dysregulation of hemostasis. 

-An elevation of neutrophils is observed, and the authors aim to link L. loa to hemostasis and thrombosis. A discussion on the possible involvement of neutrophils extracellular traps (NETs; related to their role in coagulation activation) would be in place.

PLOS authors have the option to publish the peer review history of their article (what does this mean?). If published, this will include your full peer review and any attached files.

Reviewer #1: No

Reviewer #2: No

Reviewer #3: No
---

## [Decision Letter · Decision Letter 1]

2 Dec 2024

Dear Dr. Lepage,

We are pleased to inform you that your manuscript 'Inflammation and hemostasis in loiasis before and after ivermectin treatment: a biological pilot cross-sectional study' has been provisionally accepted for publication in PLOS Neglected Tropical Diseases.

Best regards,

Feng Xue, Ph.D.

Guest Editor

Jong-Yil Chai

Section Editor

Shaden Kamhawi

co-Editor-in-Chief

Paul Brindley

co-Editor-in-Chief

Reviewer's Responses to Questions

**Key Review Criteria Required for Acceptance?**

**Methods**

-Are the objectives of the study clearly articulated with a clear testable hypothesis stated?

-Is the study design appropriate to address the stated objectives?

-Is the population clearly described and appropriate for the hypothesis being tested?

-Is the sample size sufficient to ensure adequate power to address the hypothesis being tested?

-Were correct statistical analysis used to support conclusions?

-Are there concerns about ethical or regulatory requirements being met?

Reviewer #1: (No Response)

Reviewer #2: (No Response)

Reviewer #3: (No Response)

**Results**

-Does the analysis presented match the analysis plan?

-Are the results clearly and completely presented?

-Are the figures (Tables, Images) of sufficient quality for clarity?

Reviewer #1: (No Response)

Reviewer #2: (No Response)

Reviewer #3: (No Response)

**Conclusions**

-Are the conclusions supported by the data presented?

-Are the limitations of analysis clearly described?

-Do the authors discuss how these data can be helpful to advance our understanding of the topic under study?

-Is public health relevance addressed?

Reviewer #1: (No Response)

Reviewer #2: (No Response)

Reviewer #3: (No Response)

**Editorial and Data Presentation Modifications?**

Reviewer #1: (No Response)

Reviewer #2: (No Response)

Reviewer #3: (No Response)

**Summary and General Comments**

Reviewer #1: My comments have been addressed. It is acceptable in the present form.

Reviewer #2: My concerns have been satisfactorily addressed. The manuscript now provides more clarity and scientific rigor.

Reviewer #3: The authors have addressed all my comments appropriately and have addressed my concerns.

PLOS authors have the option to publish the peer review history of their article (what does this mean?). If published, this will include your full peer review and any attached files.

Reviewer #1: No

Reviewer #2: No

Reviewer #3: No

---

## [Editor Report · Acceptance letter]

10 Dec 2024

Dear Dr. Lepage,

We are delighted to inform you that your manuscript, "Inflammation and hemostasis in loiasis before and after ivermectin treatment: a biological pilot cross-sectional study," has been formally accepted for publication in PLOS Neglected Tropical Diseases.

Best regards,

Shaden Kamhawi

co-Editor-in-Chief

Paul Brindley

co-Editor-in-Chief
